# The Machine Vision Measurement Module of the Modularized Flexible Precision Assembly Station for Assembly of Micro- and Meso-Sized Parts

**DOI:** 10.3390/mi11100918

**Published:** 2020-09-30

**Authors:** Zhiyong Zhang, Xiaodong Wang, Hongtu Zhao, Tongqun Ren, Zheng Xu, Yi Luo

**Affiliations:** 1Key Laboratory for Micro/Nano Technology and System of Liaoning Province, Dalian University of Technology, Dalian 116024, China; zzyagsd@mail.dlut.edu.cn (Z.Z.); xdwang@dlut.edu.cn (X.W.); hongtu@mail.dlut.edu.cn (H.Z.); xuzheng@dlut.edu.cn (Z.X.); luoy@dlut.edu.cn (Y.L.); 2Key Laboratory for Precision & Non-Traditional Machining of the Ministry of Education, Dalian University of Technology, Dalian 116024, China

**Keywords:** machine vision, vision system characterization, micro assembly system, precision assembly system

## Abstract

The machine vision measurement module is indispensable for the Modularized Flexible Precision Assembly Station (MFPAS), which is a fully automatic assembly system being developed at Dalian University of Technology (DUT). MFPAS consists of basic and additional modules, and are expected to be flexible, expandable, and re-configurable to adapt to a variety of parts with a large size range, requiring the machine vision measurement module to be able to achieve accurate measurement of position, as well as orientation of the parts with different size scale. An automatic zooming vision system was set up for evaluation and final integration in MFPAS. Pixel equivalent, principal point and orientation deviation of images were analyzed and experimentally studied using different magnifications of the lens. A new template with circular patterns of different diameters was designed for zoom-lens calibration. The experiments show that the measurement error caused by the variation of the pixel equivalent, principal point and orientation is estimated under 10 μm without online calibration. When high accuracy is required, online calibration can be employed during assembly. The evaluation results of the vision system with or without on-line calibration were given for a better trade-off between accuracy and efficiency during assembly.

## 1. Introduction

Assembly is one of the important and indispensable technologies for the manufacture of products. For the purpose of greater functional integration, the miniaturization of products is inevitable, and the parts to be assembled tend to be miniaturized and diversified, especially with the development of MEMS devices. The assembly accuracy is necessary to be in the range of a few micrometers. Given the limitation of human workers, developing sensor guided or sensor based automatic assembly technology is the solution to achieve high accuracy in a repeatable assembly process [1]. Among the sensors, the visual image sensor or machine vision system is undoubtedly the most important one, while other sensors including force and proximity are also utilized in assembly application.

Since the 1990s, many research efforts have been conducted on assembly methods and the development of assembly systems for micro devices or MEMS. As there are a variety of different types of small devices, various systems or machines have been developed for the assembly of different miniaturized parts. The use of vision-based feedback has been identified as one of the most promising approaches for controlling the micro assembly process [2]. As the parts to be assembled are small and specific, the task oriented vision systems utilized in most of the developed micro assembly systems have used microscopic vision with a fixed, small field of view. Some typical examples can be found in references [3,4,5,6]. When there is a requirement for vision systems with different fields of view, two or multiple vision units with different optical magnifications were employed.

As the automation of precision assembly moves forward, more and more automatic assembly machines are needed for miniaturized device manufacture. For industrial applications, vision systems are required to be more flexible for accomplishing the measurement of parts from the micro to meso scale or even larger.

The precision miniaturized devices manufactured in small and medium-sized enterprises belong to a certain type or category of products with different models, such as micro inertial sensors, bio-MEMS devices or optical modules, et al. The devices consist of a certain number of parts or components to be assembled. However, these parts or components may vary greatly in shape or size due to continuous upgrading or improvement of the devices. Meanwhile, the short R&D cycles result in the manufacturers enduring great pressure and challenges for a quick changeover. It is difficult to fulfill all the assembly tasks of a certain type or category of miniaturized devices with a dedicated assembly system developed for a specific model.

In some dedicated assembly systems developed by our research team, the problem in the application of machine vision due to the limited field of view for high resolution and a large work space was solved with a vision unit mounted on 3 DOF (degree of freedom) precision stages [7,8,9,10]. However, it is not very efficient in dealing with the parts that have different feature sizes, which may range from several micro meters to tens of millimeters. Furthermore, the adaptability to the device’s changes was limited, and the demands on the flexibility of the assembly system become much more preferred.

In order to develop more versatile assembly systems, which will be capable of accomplishing the assembly tasks of an entire category of miniaturized devices, such as a series of different models of precision accelerometers, and also be adaptable to further revisions, the Modularized Flexible Precision Assembly Station (MFPAS), a modular and more flexible automatic assembly station is being developed at DUT. It has the features of being flexible, expandable and re-configurable to meet wider assembly needs, including adhesive dispensing, micro welding with fiber laser, etc. The MFPAS can adapt to a variety of parts that need to be manipulated or measured. The architecture based on functional modularity has been taken into account in the design phase, which enables the system compactness and re-configuration. The MFPAS is expected to be developed as a modular and flexible system. It adopts the system structure of “Basic platform + functional modules”.

Many scholars have conducted in-depth research on modularity [11,12,13], which has many advantages in both design and manufacturing. Flexibility is an inevitable requirement to respond to product changes. Many efforts have been made in this aspect [14,15,16]. The flexibility of the MFPAS is mainly focused on the module for gripper exchange. A variety of exchangeable grippers already exists [17,18,19]. Unlike the existing ones, the MFPAS is equipped with hollow design structure, which facilitates the machine vision measurement module for viewing through and carrying out measurement tasks from a top view. And the machine vision measurement module can perform multi-scale measurement tasks together with the module for gripper exchange. The gripper exchange brings a high flexibility to the whole system.

With the module for gripper exchange, the system can manipulate a variety of miniaturized parts, and so therefore the system should meet the measurement requirements of the parts with a large size range by taking advantage of the machine vision module.

Compared to the fixed-parameter (passive) lens, a zoom-lens with adjustable focal length has inherent advantages in flexibility and imaging capabilities [20]. But for precision assembly, the measurement accuracy may be affected due to the change of optical magnification during the zooming process. Thus, an automatic zooming vision system was set up for evaluation and final integration in MFPAS.

The requirements of the machine vision measurement module are listed below:(1)Automatic measuring the target position and orientation of the part (through the hollow structure with no gripper attached) and the position and orientation of the part being grasped (gripper attached).(2)Automatic focusing and amplification adjustment adapting to different parts with the size range of 0.5–10 mm and the size of the finest structure is 5–10 μm, and no image stitch if possible.(3)Measurement accuracy 1.5–5.0 μm with magnification of the lens from high to low, which changes the resolution of the image from high to low; the accuracy for orientation measurement is 0.01° (the length of the part *l* = 10 mm).(4)No obvious decrease of accuracy with different amplification in single measurement procedure.(5)Automatic calibration during operation when necessary.

In general, a high magnification lens with a small field of view ensures higher measurement accuracy, but makes it difficult to measure parts with larger sizes. When low magnification lenses with large fields of view are used to measure the parts of smaller sizes, it will inevitably reduce the measurement accuracy to some extent. Naturally, the zooming vision system will introduce some new problems, such as inaccurate value of the magnification during zooming procedure. Considering the influence of magnification adjustment on measurement accuracy, pixel equivalent, principal point and orientation deviation of image were analyzed and experimentally studied in this paper. Calibration of pixel equivalent is the precondition and foundation of the measurement based on image. It establishes the proportional relationship between pixels and physical dimensions, and has important influence on positioning accuracy in the XY plane. For camera systems with fixed-parameter lenses, the principal point coordinates are the constituent elements of the camera’s intrinsic parameters, so the principal point can be obtained by camera calibration [21,22,23]. Nevertheless, the principal point could be determined based on the focus-of-expansion method for camera systems with zoom lenses. A new template, circular patterns of different diameters combined with straight lines for the evaluation of the machine vision module, was designed for above parameters’ calibration.

The purpose of this paper is to analyze the factors of the automatic zooming vision, which may affect the measurement accuracy in practical application, and verify the feasibility of the vision system developed for integration in the automatic assembly system MFPAS. In the following section, the overall architecture of the MFPAS is introduced, and the machine vision measurement module and the module for gripper exchange are described briefly. In Section 3, evaluation of the machine vision measurement module and its experimental testing results are given based on the new template. Finally, the paper is summarized with conclusions in Section 4.

## 2. Design of the Modularized Flexible Precision Assembly Station (MFPAS)

The overall design of the MFPAS has the objective of maximizing the flexibility to meet different assembly needs of a series of models of precision miniaturized devices, which belong to a certain type or category of products, including accelerometers and other similar devices. The dimension of the parts is in the range from submillimeters to tens of millimeters, and the assembly accuracy is required to be better than 10 μm. The architecture of “Basic platform + functional modules” is applied to the system, which enables the MFPAS with the ability to quickly minimize the changes or adjustments to meet new assembly requirements.

Basic modules are mainly responsible for ordinary assembly tasks. Additional modules are designed and implemented in the framework of basic modules, and are re-configurable and expandable for special assembly requirements. Thus, the MFPAS can be divided into basic and additional modules. The constituent modules of the MFPAS are shown in Figure 1.

Figure 2 shows the layout of the MFPAS. The precise 3 DOF positioning module has three degrees of freedom (X, Y, Z) and can realize precise spatial positioning with a large working space (XYZ: 550 × 500 × 300 mm^3^). The repeatability of the X, Y axis is 2 μm, and that of the Z axis is 0.5 μm. The positioning error of XY is less than 10 μm, and Z is less than 5 μm. Since the machine vision measurement module and the base mechanism for gripper exchange are both mounted on the Z axis, the measurement task can be achieved after miniaturized parts are picked up by the module for gripper exchange. An assembly force measurement module is integrated in the base mechanism, which is mainly used for contact force control in the Z direction. 1-DOF rotary stage can realize orientation adjustment of parts to be assembled and mounting of fixtures.

### 2.1. Machine Vision Measurement Module

The machine vision measurement module can achieve accurate measurement of the position and orientation of parts with a large size range from a top view. As shown in Figure 3, it mainly consists of a CCD camera, zoom lens and light sources. Some technical parameters of the CCD camera are listed in Table 1.

The selected zoom lens is NAVITAR ZOOM6000 (Navitar, Inc, New York, NY, USA), which is a telecentric lens used to eliminate parallax between an image and the object in measurement. The zoom is a stepless type, which is continuously controlled by a motor, and the range of magnification is 0.7×–4.5×. The change in focal length of the lens allows the machine vision measurement module to have the ability to measure feature sizes from several micrometers to several millimeters, the biggest feature size can be as large as 12.5 × 9.4 mm^2^. The CCD camera and zoom lens are mounted on the Z axis, which gives it a wide measurement space as large as the working space.

Different miniaturized parts have different assembly accuracy requirements, ranging from a few microns to tens of microns. Besides the positioning errors of the XYZ guide rails and the system installation errors (including guide rail, camera), the measurement errors of the vision system also affect the assembly accuracy and should also be considered. The measurement errors are caused by several factors of the vision system, including pixel equivalent, principal point and orientation deviation of image, which should be confirmed with different magnifications of the lens. This paper focuses on these measurement errors. For large measurement errors, online calibration or a necessary compensation strategy should be taken according to the specific parts assembly accuracy requirements.

### 2.2. Modules for Gripper Exchange

Modules for gripper exchange of the MFPAS consists of the base mechanism for connection and exchangeable tool magazine. The base mechanism for connection adopts a hollow design that facilitates the observation of the machine vision measurement module through it, as well as simplifies the system structure. As shown in Figure 4, it is also mounted on the Z axis, which gives it a wide operating space.

An exchangeable tool magazine is shown in Figure 5, where a different gripper or assembly tool heads are positioned by using the locating pins and holes in the magazine. The exchangeable tool magazine has two degrees of freedom (*x*-axis translation and ɑ rotation), and a 3-DOF base mechanism for connection can automatically pick and place different tool heads from the magazine. Thus, it is more flexible to perform various assembly tasks. In addition, multiple grippers or assembly tool head positions are reserved to enhance the expandability.

Different tool heads can be exchanged by using electromagnets for connection and disconnection with the base mechanism. This electromagnet has the advantages of simple control, high reliability and low power consumption. The magnetic force disappears only when the voltage of certain value is supplied, otherwise the magnetic attraction force exists. Therefore, we can easily control the electrical signal to exchange different tool heads. Moreover, the characteristic of this electromagnet ensures that the tool head will not be released even in the case of unexpected power loss. Figure 6 shows the schematic diagram of picking a tool head by the base mechanism for connection. After a specific tool head is selected, the parts are picked up. The accurate measurement of the position and orientation of the parts can be achieved from a top view via the machine vision measurement module.

## 3. The Evaluation of Machine Vision Measurement Module

### 3.1. Design of Calibration Template

A new calibration template is designed for calibration of a zoom-lens camera. As shown in Figure 7, the template has the size of 15 × 15 mm^2^, and consists of three straight lines and a circular pattern with different diameters. The circular patterns are centrally symmetric with different diameters (0.2 mm, 0.4 mm, 0.6 mm, 0.8 mm,1.0 mm and 1.2 mm), and they are designed for calculating pixels equivalent at a certain magnification. The distance of the two smallest circles near the center is 1.0 mm, and the distance between the edges of either two adjacent circles is 0.2 mm. Straight lines are used to calibrate principal point and orientation deviation of the image considering magnification changes. The width of the straight lines is 0.05 mm, the length of the vertical lines is 5 mm and that of the horizontal lines is 13 mm.

The reason for choosing the circular pattern to calculate the pixel equivalent is that the lighting intensity has an impact on the corner extraction accuracy [24]. The radius of the circle may vary with a variation of light intensity and poor focusing, but the center of the circle is basically unchanged due to the circle being centrally symmetrical, and the edge shrinks or expands simultaneously. Therefore, we selected the circular pattern to calibrate pixel equivalent value at a certain magnification.

Compared to the same diameter circular pattern, the circular pattern with different diameters is more suitable for a different field of view, i.e., the small circle is detected in the small field of view and the large circle is detected in the large field of view. As an illustration, Figure 8 shows the field of view with optical magnifications 1.0× and 4.0×. The circular pattern with different diameters avoids the circles, and are too small in a large field of view.

### 3.2. Calibration Principle

Three parameters are used for the calibration in the magnification adjustment process, i.e., pixel equivalent (spatial resolution of image scale), principal point and orientation deviation of image. Some symbols are defined in Table 2.

Pixel equivalent is an important measurement parameter for visual system and it critically affects the final measurement accuracy [25]. Pixel equivalent value under different magnifications is different; it is defined as follows:(1)$$P_{e}=L/N$$
where *L* is the actual physical size, and *N* represents the corresponding average number of pixels. The calibration process of pixel equivalent is summarized as follows:(1)Machine vision measurement module moves to the position above the template, the center of the template is roughly located at the center of the field of view;(2)The auto zoom motor moves to *M*s, auto-focus and captures the clear image of the template;(3)Detect and extract the center coordinates of the two outermost circular patterns on the template image, denoted as A(*u*_1_, *v*_1_), B(*u*_2_, *v*_2_);(4)Due to the center distance and radius of the circular pattern being known, *L* is also known. Pixel equivalent can be calculated by (1).
(2)$$N=\sqrt{{\left(u_{2}-u_{1}\right)}^{2}+{\left(v_{2}-v_{1}\right)}^{2}}$$

For the principal point, the focus-of-expansion method is employed [20]. We assume that *P*_1_(*x*_1_, *y*_1_) and *P*_2_(*x*_2_, *y*_2_) are pixel coordinates of point *P* at different magnifications. Then, linear equation connecting point *P*_1_ and *P*_2_ satisfies the following relationship according to pin-hole model theory [26]:(3)$$u_{0}(y_{1}-y_{2})+v_{0}(x_{2}-x_{1})=x_{2}y_{1}-x_{1}y_{2}$$

We can conclude that the straight line connecting the corresponding pixel coordinates at different magnifications passes through the principal point. Therefore, principal point can be obtained based on the focus-of-expansion method [20]. The intersection of any two straight lines on the template image can be used to determine the principal point. The longest straight line on the template image is used to test orientation deviation of the image with magnification adjustment. Subsequent experiments are performed based on the new calibration template.

The following two points need to be mentioned:(1)Temperature has effects on the calibration results, and MFPAS is supposed to be installed in a clean room, and the temperature of the operation environment should be 20 ± 2 °C (under a standard ambient test condition), so the temperature was kept at 20 °C in the following experiments.(2)During the calibration procedure for testing the above mentioned three parameters, the template should be kept unmoved. As the template is fixed during the process of the experiment, all measurements at different optical magnifications were carried out relative to the same reference; misalignment of the template has no effects.

### 3.3. Experiment and Discussion

The machine vision measurement module was set up for an experiment, as shown in Figure 9. The module includes CCD camera, the automatic zooming lens, the ring light source and a coaxial light source. The specifications of the camera and the lens are given in Section 2.1

#### 3.3.1. Pixel Equivalent

Due to the existence of the mechanical backlash of the motor driving mechanism, the positioning error of the stepping motor will directly affect the actual magnification of the zoom lens. Therefore, we studied the repeatability of pixel equivalent at certain magnifications. Four motion modes are defined as follows:

A1: Zoom from low magnification to high magnification;

A2: Zoom from high magnification to low magnification;

A3: Zoom with bidirectional motion;

A4: Return to the zero position first, and then move to the specified magnification.

Experiment results at a magnification of 3x were summarized in Table 3. According to the variation range, we can see that the pixel equivalent in A4 mode has better repeatability. The reason is that the zoom motor has better positioning accuracy or repeatability at zero position. In the actual assembly, we choose A4 to ensure high measurement accuracy, although it would lose some efficiency. Subsequent experiments were performed based on the same motion pattern A4.

When different tools heads are picked up by the base mechanism, distance between the object plane and the image plane of the vision system may change at the same magnification. Therefore, it is necessary to test the repeatability of pixel equivalent under different object distances. The object distance change was simulated by moving the Z axis in the experiment. The calibration template was fixed, H1, H2, H3 are the distances between the image plane and the object plane, and they are around working distance (WD) of the lens; the difference between H1 and H2 or H2 and H3 is 0.5 mm; the accuracy of movement between the heights is determined by accuracy of the Z axis, which is less than 5 μm. At each height, auto focus was performed. The experiment results are shown in Table 4.

From Table 4, we can conclude that the pixel equivalent has good repeatability under different object distances. The maximum difference of pixel equivalent was 0.003 μm/pixel. It means that the measurement error will not exceed 3 μm when *N* is less than 1000 pixels. It also proves that the focus function has good repeatability.

#### 3.3.2. Principal Point

During the zooming process, the motion of the lens group controlled by the motor may cause the optical axis to change its position and orientation. The swing of the optical axis will then change the position of the principal point, resulting in an error in image measurement between different magnifications. So it is necessary to determine principal point offset and quantify its impact on measurement accuracy. The template image was captured at different magnifications (1×, 2×, 3×, 4×). Figure 10 shows the captured template image at 2× magnification. Point *C*_1_ and *C*_2_ are the intersection of the straight lines on the template image; the linear equation of the same object point at different magnifications is fitted by least squares method, as shown in Figure 10; the unit of the coordinate of the points are pixels. The principal point can be obtained by calculating the intersection of two fitted straight lines, i.e., point O in Figure 11.

In order to test the calibration error of the principal point, the repetitive experiment was carried out for the template with different postures. Posture 1, 2 and 3 indicate that the angle between the longest straight line on the template and U axis of the image coordinate system is 0°, 45° and 90°, respectively. The experiment results are summarized in Table 5.

From Table 4, the mean coordinate of the principal point can be obtained. The maximum offset of the principal point is less than 0.6 pixels. According to the pixel size provided by the manufacturer, the maximum error is less than 2.8 μm at 1× magnification.

#### 3.3.3. Orientation Deviation of the Image

The change in focal length of the zoom lens is achieved by adjusting the relative distance between the lens groups. Due to lens manufacturing and assembly errors, orientation of the image may change during the zooming process. The repetitive experiment was carried out for magnification adjustment. The orientation deviation was obtained by processing the images at different magnifications for the angle variation between the line of the template and axis U of the image plane. The orientation deviation is expressed with deflection angle. The experiment results with magnification changing from 3× to 4×, and vice versa, were summarized in Table 6.

Through a lot of testing experiments, it is certain that the angle variation is within 0.005°. The direction of variation is uncertain, which indicates it is a random error. Moreover, the small angle variation reflects that the lens distortion is very small, especially the tangential distortion.

## 4. Conclusions

This paper presented the overall design framework of the MFPAS being developed. MFPAS is a modular and flexible assembly system, and the flexibility, expandability and re-configuration will be able to satisfy the assembly requirement of miniaturized devices, which may come from different industrial fields.

As the basic module and key part of MFPAS, the machine vision measurement module is required to adapt to the measurement of parts from micro to meso scale parts. The vision system should have a different field of view with adjustable optical magnifications, and the factors affect the measurement accuracy when magnification changes were analyzed and experimentally studied.

The machine vision measurement module with zoom lens was set up for evaluation and final integration in the MFPAS; the main results are summarized as follows:(1)Based on the repeatability of pixel equivalent, the motion mode A4 (Return to the zero position first, and then move to the specified magnification.) was chosen to ensure high measurement accuracy although it will lose some efficiency. Due to the good repeatability under different object distances, the pixel equivalent value can be used without on-line calibration.(2)The principal point offset is less than 0.6 pixels, so the maximum measurement error is less than 2.8 μm at 1× magnification. Therefore, principal point can be considered to remain unchanged during the zooming process. For the orientation deviation of the image, the angle variation is within 0.005° and its direction is random, so the measurement error could be negligible for assembly with accuracy requirements around tens of micrometers.(3)When the much higher accuracy was required during assembly, the on-line calibration should be employed.

## Figures and Tables

**Figure 1 micromachines-11-00918-f001:**
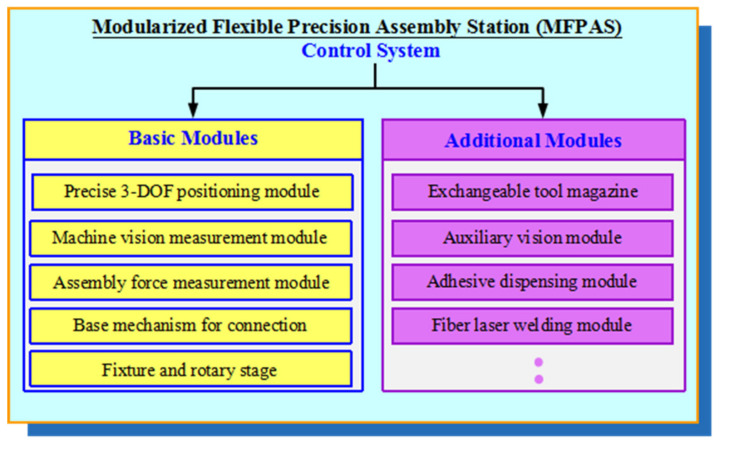
Division of modules of the Modularized Flexible Precision Assembly Station (MFPAS).

**Figure 2 micromachines-11-00918-f002:**
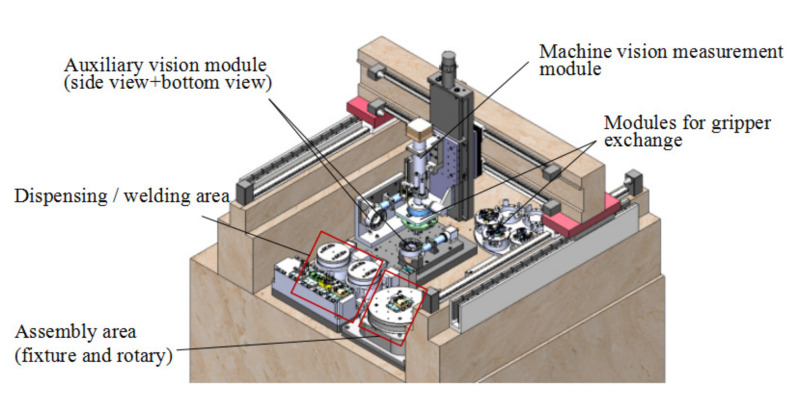
Layout of the MFPAS.

**Figure 3 micromachines-11-00918-f003:**
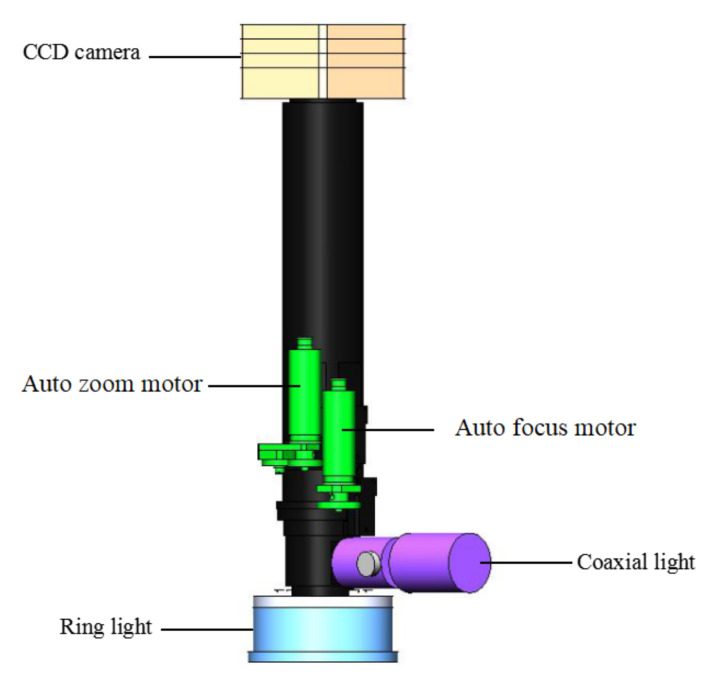
Machine vision measurement module.

**Figure 4 micromachines-11-00918-f004:**
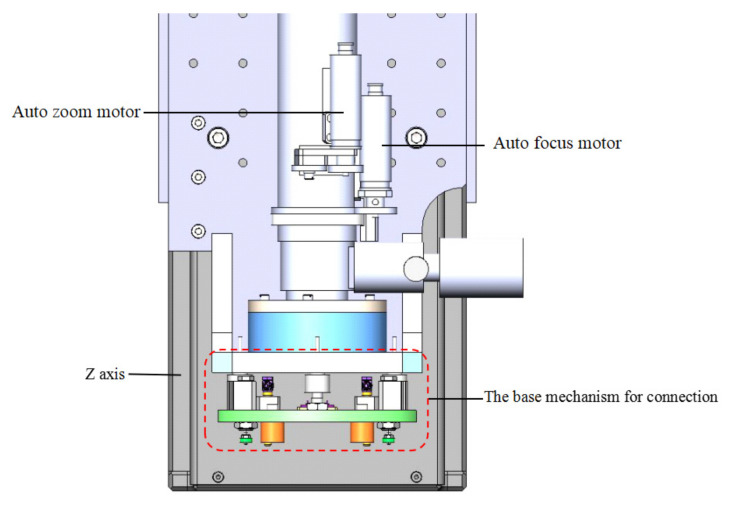
The base mechanism for connection integrated on Z axis.

**Figure 5 micromachines-11-00918-f005:**
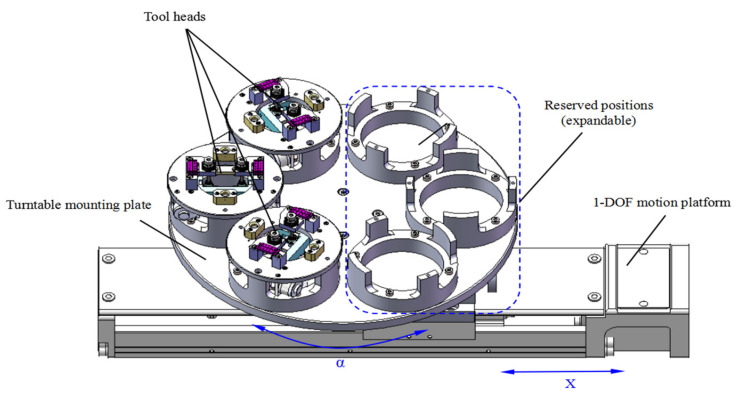
Exchangeable tool magazine.

**Figure 6 micromachines-11-00918-f006:**
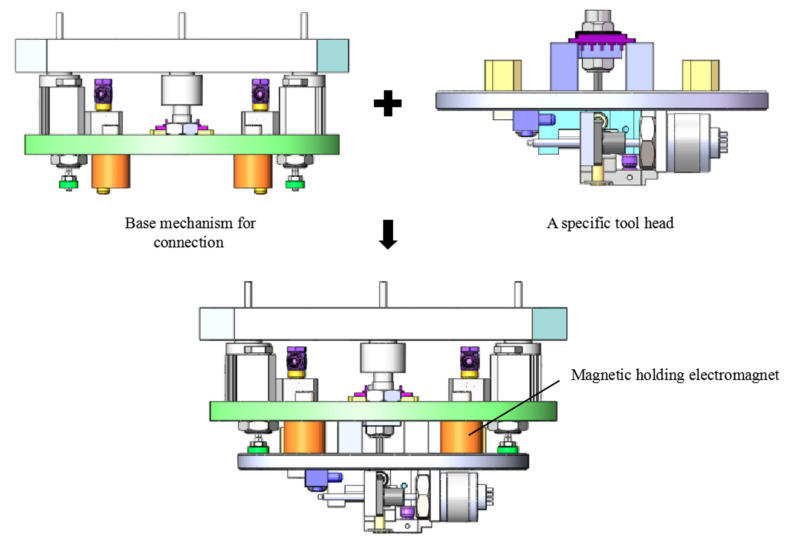
Schematic diagram of picking a specific tool head.

**Figure 7 micromachines-11-00918-f007:**
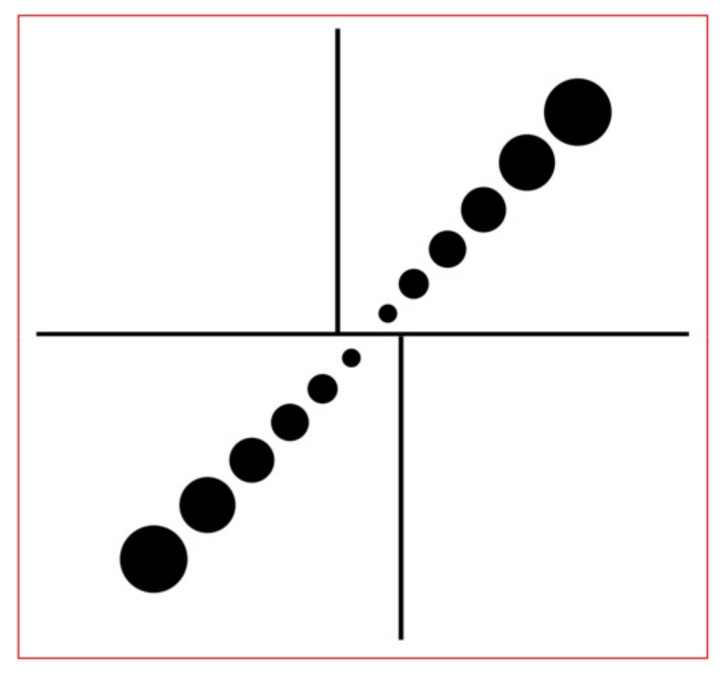
The new calibration template.

**Figure 8 micromachines-11-00918-f008:**
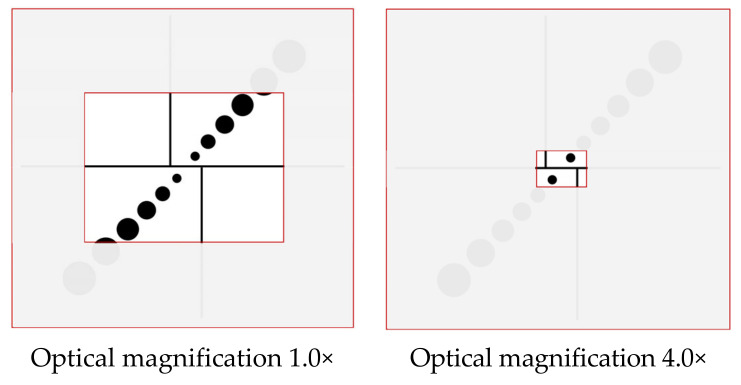
The field of view with different optical magnification.

**Figure 9 micromachines-11-00918-f009:**
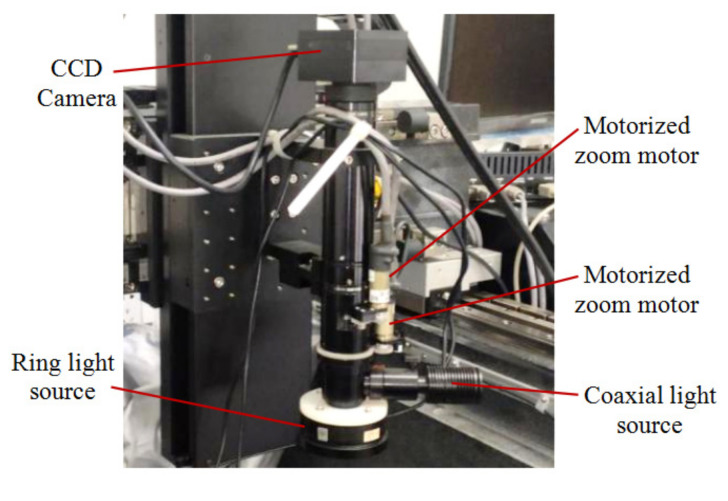
The machine vision measurement module setup for experiment.

**Figure 10 micromachines-11-00918-f010:**
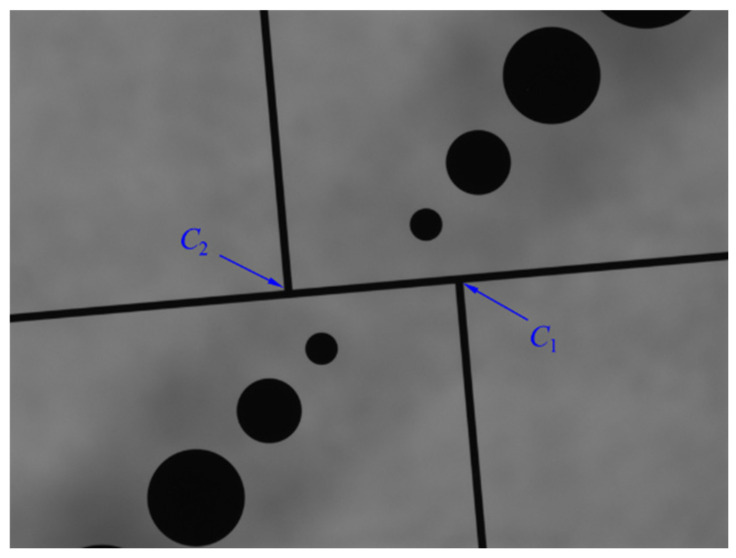
The captured template image at 2× magnification.

**Figure 11 micromachines-11-00918-f011:**
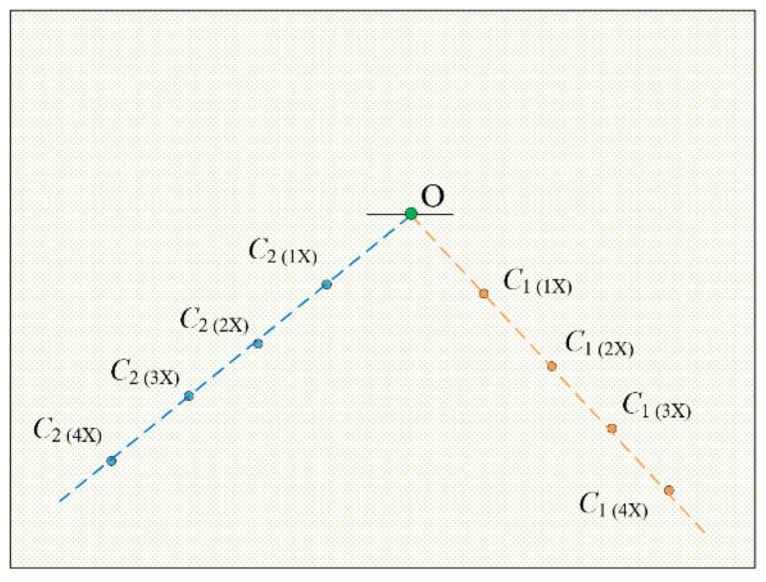
Illustration of finding the principal point O.

**Table 1 micromachines-11-00918-t001:** Some technical parameters of the CCD camera.

Version	XIMEA MD028MU-SY
Resolution	1940 × 1460
Sensor type	CCD B/W
Sensor model	Sony ICX674 ALG
Sensor size	11 mm
Sensor active area	8.8 × 6.6 mm^2^
Pixel size	4.54 µm
Dimensions *W* × *H* × *D*	60 × 60 × 38 mm^3^

**Table 2 micromachines-11-00918-t002:** Definition of some symbols.

Symbol	Definition	Units
*P* _e_	spatial resolution of image scale	μm/pixel
*u*_0_, *v*_0_	pixel coordinate values of the principal point	pixel
*Ms*	specified magnification	none
*u*, *v*	center pixel coordinate values of outermost circular pattern on the template image	pixel
*x*, *y*	pixel coordinate values of the object point *P*	pixel

**Table 3 micromachines-11-00918-t003:** The pixel equivalent in different motion modes (μm/pixel).

Experiment Number	A1	A2	A3	A4
No. 1	1.507	1.491	1.457	1.513
No. 2	1.504	1.492	1.452	1.514
No. 3	1.503	1.489	1.451	1.513
No. 4	1.505	1.491	1.455	1.512
No. 5	1.503	1.492	1.454	1.514
Maximum variation	0.004	0.003	0.006	0.002

**Table 4 micromachines-11-00918-t004:** The pixel equivalent under different object distances (μm/pixel).

Magnification	H1	H2	H3	Maximum Variation
2×	2.279	2.279	2.279	0.003
	2.282	2.280	2.279	
	2.279	2.280	2.279	
3×	1.513	1.516	1.516	0.003
	1.514	1.516	1.513	
	1.516	1.514	1.514	
4×	1.137	1.138	1.137	0.001
	1.138	1.138	1.137	
	1.137	1.137	1.137	

**Table 5 micromachines-11-00918-t005:** The coordinates of the principal point in different postures of calibration template (pixel).

Experiment Number	Posture 1	Posture 2	Posture 3
No. 1	(917.048, 697.517)	(917.486, 697.273)	(917.385, 697.177)
No. 2	(917.135, 697.308)	(917.392, 697.314)	(917.030, 697.140)
No. 3	(917.208, 697.408)	(917.500, 697.293)	(917.060, 697.377)
No. 4	(917.183, 697.369)	(917.503, 697.285)	(917.140, 697.243)
No. 5	(917.040, 697.446)	(917.466, 697.382)	(917.150, 697.330)
Mean value	(917.2484, 697.3241)		
Maximum deviation	Δ*u*: 0.473, Δ*v*: 0.377, Δ*d* = (Δ*u*^2^ + Δ*v*^2^)^1/2^ ≈ 0.6		

**Table 6 micromachines-11-00918-t006:** The deflection angle from 3× to 4×.

Experiment Number	Angle Variation (°)
No. 1	−0.00365
No. 2	−0.00153
No. 3	−0.00235
No. 4	0.00091
No. 5	−0.00146
Mean value	−0.001616
Standard uncertainty	0.000745

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
