# Peer review of "The Machine Vision Measurement Module of the Modularized Flexible Precision Assembly Station for Assembly of Micro- and Meso-Sized Parts"

_micromachines, 2020, doi:10.3390/mi11100918_

Round 1

Reviewer 1 Report

This paper reported a machine vision platform for assembly of miniature parts. Overall, it is difficult to understand because of the poor English, and the paper requires extensive English language editing. There is lacking experimental results to verify the claim that the platform is flexible for assembly of parts with different shapes and sizes. Without experimental verification, it is not clear if the proposed machine vision platform is actually working or not. Although the authors presented the results of pixels equivalent to different objective distances, it needs further discussions and verifications about how it can improve the flexibility and why it can be adaptable to parts with different shapes and sizes.

Reviewer 2 Report

Review of the paper entitled

“The machine vision measurement module of the modularized flexible precision assembly station (MFPAS) for assembly of micro to medium sized parts”

General considerations

The paper is quite clear in the approach and methods.  It refers an interesting integrated vision system for metrological characterization and micro-meso parts assembly. Conclusions are consistent with experimental results. Results and the developed knowledge are useful for the scientific community. However, the clearness of the paper should be improved. Some details of the setup are required in order to save the reproducibility of this work. Some results are not adequately commented or justified. In addition, in some sections, a more accurate description would be appreciated. Some sentence are not clear and need to be revised. 

Some major comments:

  1. Title should be improved. For example:
    • Do not use acronyms in title, unless: (a) the subject is almost exclusively known by its acronym or is widely known and used in that form, and (b) the acronym does not commonly have more than one expansion. (Acronyms that can be used in titles include MEMS, UV, 2-D, 3-D, DNA, etc.). Therefor (MFPAS) should be avoided.
    • “Meso” is more proper than “medium”.
    • Title should be more generic, not strictly connected to a specific assembly system. Is the machine vision measurement module applicable to other (micro/meso-)assembly machine?
  2. Affiliations should appear with the same order or authors. First author should have affiliation reference 1, and so on. Some authors have two affiliations. Is that correct?
  3. Abstract.
    • Acronyms (i.e. MFPAS, DUT) should be defined in the abstract.
    • As suggested in the “instructions for authors” document (See https://www.mdpi.com/journal/micromachines/instructions), the abstract should briefly summarize results of the research. It is suggested to add some results.
  4. Keywords are not adequate. Please, revise the template of the “Micromachines” for instructions. See mdpi.com/files/word-templates/micromachines-template.dot
    • Keywords section should report technologies, fields of knowledge, methods, tools, etc. which are matters treated into the paper.
    • As in titles, rules of acronyms. MFPAS is a specific and custom acronym, which should be avoided.
    • “List three to ten pertinent keywords specific to the article; yet reasonably common within the subject discipline”.
    • Examples for authors: “Vision system” is allowed. “Zoom vision system” is not allowed; “pixel equivalent” is not allowed. “principal point” is not allowed, “Vision system calibration” is allowed. “MFPAS” is not allowed, “micro-assembly system” or “assembly systems” are allowed.
  5. Section 1 – Introduction. The state-of-the-Art is too focused on the MFPAS. Other general-purpose assembly machine and vision systems have been proposed by other authors. What limits the available systems? Comparison between the MFPAS and other available systems? How these limits can be overcome with the proposed solution?
  6. Section 1, Page 2, row 62. A reference is required. How the requirements are defined/identified?
  7. Section 1, Page 2, rows 67-68. Check units.
  8. Section 1, Page 2, rows 68-69. This sentence is ambiguous. If the accuracy for orientation measurements is 0.01° how authors can write (1=10mm)? What does it mean? Please, check this requirement and clarify.
  9. [14] seems to be “forced” or not so much consistent. Ref. [14] is referred to a specific application case.
  10. Section 2 – Design of the MFPAS. Page 3, Rows 98-99. Too generic. Authors should better explain accuracy limits and parts dimensions assembled by the proposed system.
  11. Section 2 – Design of the MFPAS. Page 3, Rows 108-109. As comment #10.
  12. Section 2, page 3 rows 108-115. More information about the assembly system are required. For example, axis strokes? Accuracy in positioning? Repeatability? Overall working volume? Especially for z-axis.
  13. Section 2.1, page 4, rows 128-129. As comment #10 and #11, there is a lack of information.
  14. Page 4 row 126. “Range of magnification is 0.74.5.” Please, check and correct.
  15. Figure 3 and figure 4. Expression as “motorized zoom motor” or “motorized focus motor” are not proper.
  16. Section 3.1, page 6, rows 167-169. The parameters are not defined clearly. In addition, the name of the parameter “Pixel equivalent” seems to be not in accordance with the state-of-the-art, where it is referred as “spatial resolution” of “image scale”. Also “principal point” seems to be not aligned to literature nomenclature, where it is referred as “center point” or “base point” of the image.
  17. Section 3.1, page 7, Figure 8 is not acceptable. Since it is a list of parameters, they should be put into a table, specifying: i. Symbol; ii. Definition; iii. Units.
  18. “Ms” is the set point of the magnification, i.e. the target value of magnification, but this is not clear in the paper, except for page 7 row 190, where it is stated that the zoom is moved to reach the Ms.
  19. Section 3.2, page 7, rows 197-198. A reference for pin-hole model theory is required.
  20. Section 3.2, page 8, rows 202-204. It is suggested to authors to better clarify the method.
  21. Section 3.3.1, page 8, row 220. Do authors intend that the Ms=3 ? If so, please, improve clearness.
  22. Section 3.3.1, page 8, row 222. Readers would appreciate a definition of the “standard uncertainty”. It seems to be an estimation of the standard deviation of the measurements but an explicit definition or declaration would improve the clearness.
  23. Table 3. Distances H1, H2 and H3 should be better specified. Units? How these values are measured? What is the accuracy of these distance measurements?
  24. Section 3.3.1, page 8, row 238. About the value 0.003, units should be always specified after each measurements. No matter it is clear from the contest.
  25. Section 3.3.1, page 9, rows 248-249. How calculations of mean least squared method fitting is performed? Algorithm? Authors can evaluate the opportunity to add a graph to improve the comprehension of the method.
  26. Section 3.3.2, Page 10, row 254. Please define “posture”.
  27. Section 3.3.2, Page 10, row 257. The maximum offset of the measured principal point is 0.6 pixels but the maximum deviation is the variation of u0, which is 0.473 pixels.
  28. Section 3.3.2, Page 10, row 262. More details of how the experiments have been execute are required. No information have been reported about the angle measurements.
  29. Section 3.3.2, Page 10, rows 265-267. Which “testing experiments”? 0.005°?
  30. Section 4 – Conclusions. Page 10, rows
  31. Section 4, page 10, rows 271-272. As comment #10.”Certain type or category” is too undefined. This is not allowed.
  32. Section 4, page 11, rows 281-282. What is “general assembly requirements”? Authors cannot

Other minor comments:

  1. In order to increase the clarity, authors should revise some sentences. Some examples:
    • at page 1, rows 28-30. The meaning of the sentence is not clear: authors are suggested to avoid too generic expressions like “certain type or category”. This expression recurs in several point in all the paper. In addition, some terms are not proper: “miniature” instead of “miniaturized”, or “in small and medium sized” instead of “at micro and meso-scale”.
    • In the abstract, sentence at rows 22-24: “When high accuracy requirement is required…”
    • Page 1 row 41, multiple “or wider” assembly needs? Or both?
    • Page 2, row 95. “conclusions” seems to be more appropriate than “concludes”.
    • Section 2.1, page 4 rows 125-126. Please, revise the sentence for more clarity.
    • Section 2.1, page 4 rows 131-134. Please, revise the sentence for more clarity.
    • Section 2.2 page 5, rows 151-155. The expression “magnetic holding electromagnet” is not proper. It is suggested to revise it.
    • Section 2.2 page 5, rows 153-154. A revision is required for more clarity.
    • Section 2.2 page 6, rows 158. A revision is required for more clarity.
    • Section 3.1 page 7, rows 172-174. A revision is required for more clarity.
    • Section 3.3.1, page 9, rows 229-231. It is suggested to revise the sentence.
    • Section 3.3.1, page 9, rows 233-234. It is suggested to revise the sentence.
    • Section 3.3.1, page 9, rows 242-244. Please, revise the sentence.
    • Section 3.3.2, Page 10, row 265. “Deflection” seems to be improper within the sentence.

  1. The paper must be improved also in terms of formatting and compliancy with the template. Some remarks:
    • Table 3 can be improved by introducing a horizontal grid and distinguishing the three values of magnifications.
    • Table 3. The header Pe of “mean value” and standard uncertainty” should be specified, for more clarity.
    • Figure 10. Labels “C1” and “C2” are too small. Increase the dimensions to let them more visible.
  2. Typos: Page 3, row 99. A full stop “.” is needed.

Round 2

Reviewer 2 Report

All comments have been satisfactory addressed and the paper has been significantly improved in all sections. The State-of-the-Art is now well written and all information are correctly reported. Solutions, methods and experimental tests are now described adequately. Results discussion and conclusions have been improved too. The readability in increased significantly.

Author Response

Thank you for your suggestion.